# Exploration via Elliptical Episodic Bonuses

**Mikael Henaff**
Meta AI Research
mikaelhenaff@meta.com

**Roberta Raileanu**
Meta AI Research
raileanu@meta.com

**Minqi Jiang**
University College London
Meta AI Research
meta@fb.com

**Tim Rocktäschel**
University College London
t.rocktaschel@cs.ucl.ac.uk

## Abstract

In recent years, a number of reinforcement learning (RL) methods have been proposed to explore complex environments which differ across episodes. In this work, we show that the effectiveness of these methods critically relies on a count-based episodic term in their exploration bonus. As a result, despite their success in relatively simple, noise-free settings, these methods fall short in more realistic scenarios where the state space is vast and prone to noise. To address this limitation, we introduce **E**xploration via **E**lliptical **E**pisodic **B**onuses (**E3B**), a new method which extends count-based episodic bonuses to continuous state spaces and encourages an agent to explore states that are diverse under a learned embedding within each episode. The embedding is learned using an inverse dynamics model in order to capture controllable aspects of the environment. Our method sets a new state-of-the-art across 16 challenging tasks from the MiniHack suite, without requiring task-specific inductive biases. E3B also matches existing methods on sparse reward, pixel-based Vizdoom environments, and outperforms existing methods in reward-free exploration on Habitat, demonstrating that it can scale to high-dimensional pixel-based observations and realistic environments.

## 1 Introduction

Exploration in environments with sparse rewards is a fundamental challenge in reinforcement learning (RL). In the tabular setting, provably optimal algorithms have existed since the early 2000s [36, 10]. More recently, exploration has been studied in the context of deep RL, and a number of empirically successful methods have been proposed, such as pseudocounts [9], intrinsic curiosity modules (ICM) [52], and random network distillation (RND) [13]. These methods rely on intrinsically generated exploration bonuses that reward the agent for visiting states that are novel according to some measure. Different measures of novelty have been proposed, such as the likelihood of a state under a learned density model, the error of a forward dynamics model, or the loss on a random prediction task. These approaches have proven effective on hard exploration problems, as exemplified by the Atari games Montezuma's Revenge and PitFall [22].

The approaches above are, however, designed for *singleton* RL tasks, where the agent is spawned in the same environment in every episode. Recently, several studies have drawn attention to the fact that RL agents exhibit poor generalization across environments, and that even minor changes to the environment can lead to substantial degradation in performance [75, 35, 74, 18, 38]. This has motivated the creation of benchmarks in the Contextual Markov Decision Process (CMDP) framework, where different episodes correspond to different environments that nevertheless share certain characteristics. Examples of CMDPs include procedurally generated (PCG) environments

[15, 58, 41, 34, 17, 7, 29, 53] or embodied AI tasks where the agent must generalize its behavior to unseen physical spaces at test time [59, 61, 28, 72]. A number of methods have been proposed which have shown promising performance in PCG environments with sparse rewards, such as RIDE [56], AGAC [25] and NovelD [76]. These methods propose different intrinsic reward functions, such as the change in representation in a latent space, the divergence between the predictions of a policy and an adversary, or the difference between random network prediction errors at two consecutive states. Although not presented as a central algorithmic feature, these methods also include a count-based bonus which is computed at the episode level.

In this work, we take a closer look at exploration in CMDPs, where each episode corresponds to a different environment context. We first show that, surprisingly, the count-based episodic bonus that is often included as a heuristic is in fact essential for good performance, and current methods fail if it is omitted. Furthermore, due to this dependence on a count-based term, existing methods fail on more complex tasks with irrelevant features or dynamic entities, where each observation is rarely seen more than once. We find that performance can be improved by counting certain features extracted from the observations, rather than the observations themselves. However, different features are useful for different tasks, making it difficult to design a feature extractor that performs well across all tasks.

To address this fundamental limitation, we propose a new method, E3B, which uses an elliptical bonus [6, 20, 42] at the episode level that can be seen as a natural generalization of a count-based episodic bonus to continuous state spaces, and that is paired with a self-supervised feature learning method using an inverse dynamics model. Our algorithm is simple to implement, scalable to large or infinite state spaces, and achieves state-of-the-art performance across 16 challenging tasks from the MiniHack suite [58], without the need for task-specific prior knowledge. It also matches existing methods on hard exploration tasks from the VizDoom environment [37], and significantly outperforms existing methods in reward-free exploration on the Habitat embodied AI environment [59, 69]. This demonstrates that E3B scales to rich, high dimensional pixel-based observations and real-world scenarios. Our code is available at https://github.com/facebookresearch/e3b.

## 2 Background

### 2.1 Contextual MDPs

We consider a Contextual Markov Decision Process [1] (CMDP [30]) given by $\mathcal{M} = (\mathcal{S}, \mathcal{A}, \mathcal{C}, P, r, \mu_C, \mu_S)$ where $\mathcal{S}$ is the state space, $\mathcal{A}$ is the action space, $\mathcal{C}$ is the context space, $P$ is the transition function, $r$ is the reward function, $\mu_C$ is the distribution over contexts and $\mu_S$ is the conditional initial state distribution. At each episode, we sample a context $c \sim \mu_C$, an initial state $s_0 \sim \mu(\cdot|c)$, and subsequent states in the episode are sampled according to $s_{t+1} \sim P(\cdot|s_t, a_t, c)$. Let $d_\pi^c$ denote the distribution over states induced by following policy $\pi$ in context $c$. The goal is to learn a policy $\pi$ which maximizes the expected return over all contexts, i.e. $R = \mathbb{E}_{c \sim \mu_C, s \sim d_\pi^c, a \sim \pi(s)}[r(s, a)]$. Examples of CMDPs include procedurally-generated environments, such as ProcGen [17], MiniGrid [15], NetHack [41], or MiniHack [58], where each context $c$ corresponds to the random seed used to generate the environment; in this case, the number of contexts $|\mathcal{C}|$ is effectively infinite. Other examples include embodied AI environments [59, 69, 28, 61, 72], where the agent is placed in different simulated houses and must navigate to a location or find an object. In this setting, each context $c \in \mathcal{C}$ represents a house identifier and the number of houses $|\mathcal{C}|$ is typically between 20 and 1000. For an in-depth review of the literature on CMDPs and generalization in RL, see [39].

### 2.2 Exploration Bonuses

If the environment rewards are sparse, learning a policy using simple $\epsilon$-greedy exploration may require intractably many samples. We therefore consider methods that augment the external reward function $r$ with an intrinsic reward bonus $b$. A number of intrinsic bonuses that encourage exploration in singleton (non-contextual) MDPs have been proposed, including pseudocounts [9], intrinsic curiosity modules (ICM) [52] and random network distillation (RND) error [13]. At a high level, these methods

---

[1]Technically, some of the environments we consider are Contextual Partially Observed MDPs (CPOMDPs), but we follow the convention in [39] and adopt the CMDP framework for simplicity. For CPOMDPs, we use recurrent networks or frame stacking to convert to CMDPs, as done in prior work [46].

define an intrinsic reward bonus that is high if the current state is different from the previous states visited by the agent, and low if it is similar (according to some measure).

| Method | Exploration bonus |
| --- | --- |
| RIDE [56] | $\|\phi(s_{t+1}) - \phi(s_t)\|_2 \cdot 1/\sqrt{N_e(s_{t+1})}$ |
| NovelD [76] | $\left[b_{\mathrm{RND}}(s_{t+1}) - \alpha \cdot b_{\mathrm{RND}}(s_t)\right]_+ \cdot \mathbb{I}[N_e(s_{t+1}) = 1]$ |
| AGAC [25] | $D_{\mathrm{KL}}(\pi(\cdot|s_t)\|\pi_{\mathrm{adv}}(\cdot|s_t)) + \beta \cdot 1/\sqrt{N_e(s_{t+1})}$ |

Table 1: Summary of recent exploration methods for procedurally-generated environments. Each exploration bonus has a count-based episodic term, marked in blue.

More recently, several methods have been proposed for and evaluated on procedurally-generated MDPs. All use different exploration bonuses, which are summarized in Table 1. RIDE [56] defines a bonus based on the distance between the embeddings of two consecutive observations, NovelD [76] uses a bonus based on the difference between two consecutive RND bonuses, and AGAC [25] uses the KL divergence between the predictions of the agent's policy and those of an adversarial policy trained to mimic it (see the original works for details). In addition, all three methods have a term in their bonus (marked in blue) which depends on $N_e(s_{t+1})$ – the number of times $s_{t+1}$ has been encountered during the *current episode*. Although presented as a heuristic, below we will show that without this count-based episodic term, all three methods fail to learn. This in turn limits their effectiveness in more complex, dynamic, and noisy environments.

## 3 Importance and Limitations of Count-Based Episodic Bonuses

We now discuss in more detail the importance and limitations of the count-based episodic bonuses used in RIDE, AGAC and NovelD, which depend on $N_e(s_t)$. Figure 1 shows results for the three methods with and without their respective count-based episodic terms, on one of the MiniGrid environments used in prior work. When the count-based terms are removed, all three methods fail to learn. Similar trends apply for other MiniGrid environments (see Appendix D.1). This shows that the episodic bonus is in fact essential for good performance.

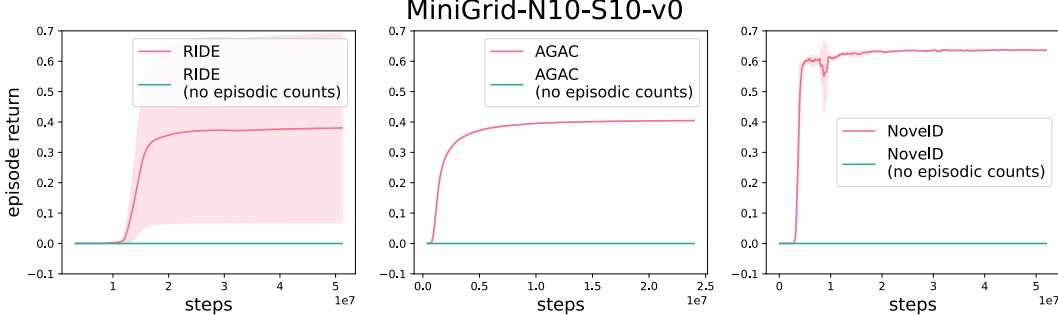

Figure 1: Results for RIDE, AGAC and NovelD with and without the count-based episodic bonus, over 5 random seeds (solid line indicates the mean, shaded region indicates one standard deviation). All three algorithms fail if the count-based episodic bonus is removed.

However, the count-based episodic bonus suffers from a fundamental limitation, which is similar to that faced by count-based approaches in general: if each state is unique, then $N_e(s_t)$ will always be $1$ and the episodic bonus is no longer meaningful. This is the case for many real-world applications. For example, a household robot's state as recorded by its camera might include moving trees outside the window, clocks showing the time or images on a television screen which are not relevant for its tasks, but nevertheless make each state unique.

Previous works [56, 25, 76] have used the MiniGrid test suite [15] for evaluation, where observations are less noisy and do not typically contain irrelevant information. Thus, methods relying on episodic counts have been effective in these scenarios. However, in more complex environments such as

MiniHack [58] or with high-dimensional pixel-based observations, episodic count-based approaches can cease to be viable.

A possible alternative could be to design a function to extract relevant features from each state and feed them to the count-based episodic bonus. Specifically, instead of defining a bonus using $N_e(s_t)$, we could define the bonus using $N_e(\phi(s_t))$, where $\phi$ is a hand-designed feature extractor. For example, in the paper introducing the MiniHack suite [58], the RIDE implementation uses $\phi(s_t) = (x_t, y_t)$, where $(x_t, y_t)$ is the spatial location of the agent at time $t$. However, this approach relies heavily on task-specific knowledge.

Figure 2 shows results on two tasks from the MiniHack suite for three NovelD variants (we decided to focus our study on variants of NovelD, since it was previously shown to outperform competing methods on MiniGrid [76]). The first, NOVELD, denotes the standard formulation which uses the bonus $\mathbb{I}[N_e(s_t) = 1]$. The second, NOVELD-POSITION, uses the positional feature encoding described above, i.e. the episodic bonus is defined as $\mathbb{I}[N_e(\phi(s_t)) = 1]$ with $\phi(s_t) = (x_t, y_t)$. The third, NOVELD-MESSAGE, uses a feature encoding where $\phi(s_t)$ extracts the message portion of the state $s_t$ similarly to [47] (both encodings are explained in more detail in Section 5).

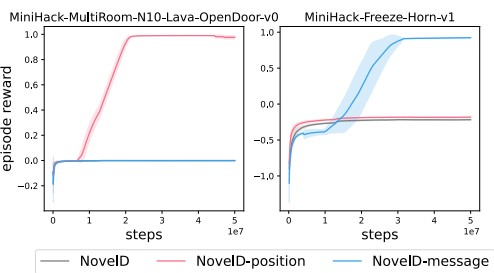

In contrast to the MiniGrid environments, here standard NOVELD fails completely due to the presence of a time counter feature in the Mini-Hack observations, which makes each observation in the episode unique. Using the positional encoding enables NOVELD-POSITION to solve the `MultiRoom` task, but this method fails on the `Freeze` task. On the other hand, using the message encoding enables NOVELD-MESSAGE to succeed on the `Freeze` task, but it fails on the `MultiRoom` one. This illustrates that different feature extractors are effective on different tasks, and that designing one which is broadly effective is challenging. Therefore, robust new methods which do not require task-specific engineering are needed.

Figure 2: Performance of NovelD with different feature extractors: entire observation, agent location, or environment message. Results are averaged over 5 seeds and the shaded region represents one standard deviation.

## 4 Elliptical Episodic Bonuses

In this section we describe **Exploration via Elliptical Episodic Bonuses**, (**E3B**), our algorithm for exploration in contextual MDPs. It is designed to address the shortcomings of count-based episodic bonuses described above, with two aims in mind. First, we would like an episodic bonus that can be used with *continuous* state representations, unlike the count-based bonus which requires discrete states. Second, we would like a representation learning method that only captures information about the environment that is relevant for the task at hand. The first requirement is met by using an elliptical bonus [6, 20, 42], which provides a continuous analog to the count-based bonus, while the second requirement is met by using a representation learned with an inverse dynamics model [52, 56].

A summary of the method is shown in Figure 3. We define an intrinsic reward based on the position of the current state's embedding with respect to an ellipse fit on the embeddings of previous states encountered within the same episode. This bonus is then combined with the environment reward and used to update the agent's policy. The next two sections describe the elliptical bonus and embedding method in detail.

### 4.1 Elliptical Episodic Bonus

Given a feature encoding $\phi$, at each time step $t$ in the episode the elliptical bonus $b$ is defined as follows:

$$b(s_t) = \phi(s_t)^\top C_{t-1}^{-1} \phi(s_t), \qquad C_{t-1} = \sum_{i=1}^{t-1} \phi(s_i)\phi(s_i)^\top + \lambda I \qquad (1)$$

Here $\lambda I$ is a regularization term to ensure that the matrix $C_{t-1}$ is non-singular, where $\lambda$ is a scalar coefficient and $I$ is the identity matrix. The reward optimized by the algorithm is then defined as

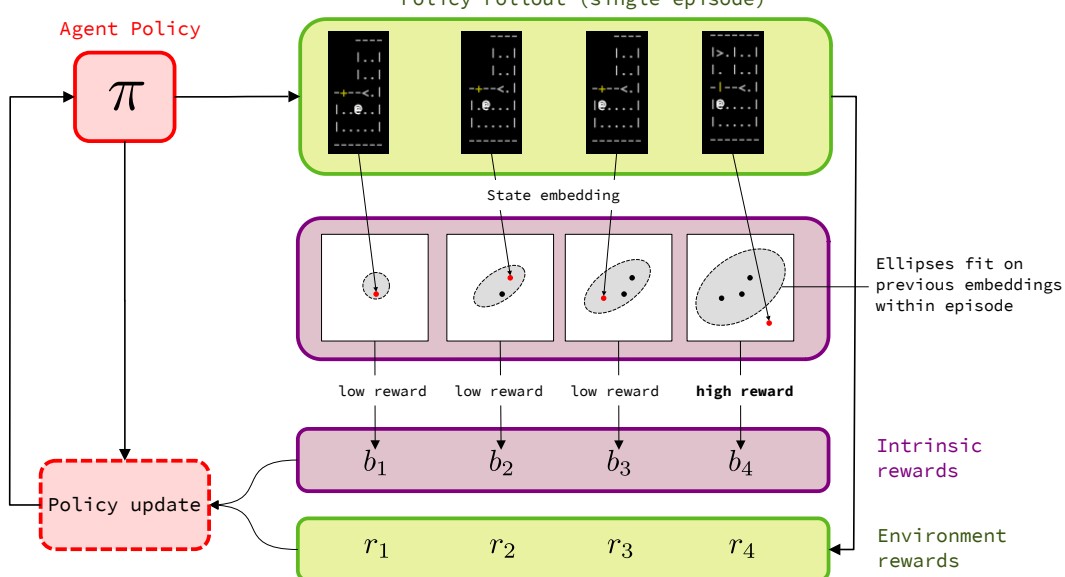

Figure 3: Overview of E3B. At each step in an episode, an ellipse is fit on the embeddings of previous states encountered within the episode. Intrinsic rewards are computed based on the position of the current state's embedding with respect to the ellipse: the further outside the ellipse, the higher the reward. These are combined with environment rewards, the policy is updated, and the process repeated. The state embeddings are learned using an inverse dynamics model.

$\bar{r}(s_t, a_t) = r(s_t, a_t) + \beta \cdot b(s_t)$, where $r(s_t, a_t)$ is the extrinsic reward provided by the environment and $\beta$ is a scalar term balancing the tradeoff between exploration and exploitation.

**Intuition.** One perspective is that the elliptical bonus is a natural generalization of a count-based episodic bonus [2]. To see this, observe that if the problem is tabular and $\phi$ is a one-hot encoding of the state, then $C_{t-1}$ will be a diagonal matrix whose entries contain the counts corresponding to each state encountered in the episode. Its inverse $C_{t-1}^{-1}$ will also be a diagonal matrix whose entries are inverse state visitation counts, and the bilinear form $\phi(s_t)^\top C_{t-1}^{-1}\phi(s_t)$ reads off the entry corresponding to the current state $s_t$, yielding a bonus of $1/N_e(s_t)$.

For a more general geometric interpretation, if $\phi(s_0), ..., \phi(s_{t-1})$ are roughly centered at zero, then $C_{t-1}$ can be viewed as their unnormalized covariance matrix. Now consider the eigendecomposition $C_{t-1} = U^\top \Lambda U$, where $\Lambda$ is the diagonal matrix whose entries are the eigenvalues $\lambda_1, ..., \lambda_n$ (these are real since $C_{t-1}$ is symmetric). Letting $z = U\phi(s_t) = (z_1, ..., z_n)$ be the set of coordinates of $\phi(s_t)$ in the eigenspace of $C_{t-1}$, we can rewrite the elliptical bonus as:

$$b(s_t) = z^\top \Lambda^{-1} z = \sum_{i=1}^{n} \frac{z_i^2}{\lambda_i}$$

The bonus increases the more $\phi(s_t)$ is aligned with the eigenvectors corresponding to smaller eigenvalues of $C_{t-1}$ (directions of low data density), and decreases the more it is aligned with eigenvectors corresponding to larger eigenvalues (directions of high data density). An illustration for $n = 2$ is shown in Figure 10 of Appendix B. In our experiments, $n$ is typically between $256$ and $1024$.

**Efficient Computation.** The matrix $C_{t-1}$ needs to be inverted at each step $t$ in the episode, an operation which is cubic in the dimension of $\phi$ and may thus be expensive. To address this, we use the Sherman-Morrison matrix identity [62] to perform fast rank-1 updates in quadratic time:

$$C_t^{-1} = \begin{cases} \frac{1}{\lambda}I & \text{if } t = 0 \\ C_{t-1}^{-1} - \frac{C_{t-1}^{-1}\phi(s_t)\phi(s_t)^\top C_{t-1}^{-1\top}}{1+\phi(s_t)^\top C_{t-1}^{-1}\phi(s_t)} & \text{if } t \geq 1 \end{cases}$$

This results in an approximately $3\times$ speedup over naïve matrix inversion (details in Appendix D.2).

## 4.2 Learned Feature Encoder

Any feature learning method could in principle be used to learn $\phi$. Here we use the inverse dynamics model approach proposed in [52], which trains a model $g$ along with $\phi$ to map each pair of consecutive embeddings $\phi(s_t), \phi(s_{t+1})$ to a distribution over actions $a_t$ linking them. In our setup, $\phi$ is separate from the policy network. The $g$ model is trained jointly with $\phi$ using the following per-sample loss:

$$\ell(s_t, a_t, s_{t+1}; \phi, g) = -\log p(a_t | g(\phi(s_t), \phi(s_{t+1})))$$

The motivation is that the mapping $\phi$ will discard information about the environment which is not useful for predicting the agent's actions. Previous work [52] has shown that this can make learning more robust to random noise or other parts of the state which are not controllable by the agent.

## 4.3 Full Algorithm

Putting all of these together, the full algorithm is given below.

---

**Algorithm 1** Exploration via Episodic Elliptical Bonuses (E3B)

---

Initialize policy $\pi$, feature encoder $\phi$ and inverse dynamics model $f$.
**while** not converged **do**
    Sample context $c \sim \mu_C$ and initial state $s_0 \sim \mu_S(\cdot|c)$
    Initialize inverse covariance matrix: $C_0^{-1} = \frac{1}{\lambda}I$
    **for** $t = 0, ..., T$ **do**
        $a_t \sim \pi(\cdot|s_t)$                           // Sample action
        $s_{t+1}, r_{t+1} \sim P(\cdot|s_t, a_t)$         // Step through environment
        $b_{t+1} = \phi(s_{t+1})^\top C_t^{-1} \phi(s_{t+1})$     // Compute bonus
        $u = C_t^{-1} \phi(s_{t+1})$
        $C_{t+1}^{-1} = C_t^{-1} - \frac{1}{1+b_{t+1}} u u^\top$     // Update inverse covariance matrix
        $\bar{r}_{t+1} = r_{t+1} + \beta b_{t+1}$
    **end for**
    Perform policy gradient update on $\pi$ using rewards $\bar{r}_1, ..., \bar{r}_T$.
    Update $\phi$ and $g$ using $\{(s_t, a_t, s_{t+1})\}_{t=0}^{T-1}$ to minimize the loss:
$$\ell = -\log(p(a_t | f(\phi(s_t), \phi(s_{t+1}))))$$
**end while**

---

# 5 Experiments

## 5.1 MiniHack Suite

In order to probe the capabilities of existing methods and evaluate E3B, we seek CMDP environments which exhibit challenges associated with realistic scenarios, such as sparse rewards, noisy or irrelevant features, and large state spaces. For our first experimental testbed, we opted for the procedurally generated tasks from the MiniHack suite [58], which is itself based on the NetHack Learning Environment [41]. NetHack is a notoriously challenging roguelike video game where the agent must navigate through procedurally generated dungeons to recover a magical amulet. The MiniHack tasks contain numerous challenges such as finding and using magical objects, navigating through levels while avoiding lava, and fighting monsters. Furthermore, rewards are sparse and as detailed below, the state representation contains a large amount of information, only some of which is relevant for a given task.

For all our experiments we use the Torchbeast [40] implementation of IMPALA [23] as our base RL algorithm. For certain skill-based tasks, we restricted the action space to the necessary actions for solving the task at hand, since we found that none of the methods were able to make progress with

the full action space (see Appendix C.1.4). See Appendix C.1.3 for environment details and C.1 for other experiment details.

### 5.1.1 Modalities for Episodic Bonus

The MiniHack environments provide an observation at each step that includes three different modalities: i) a symbolic image, which indicates the location of the agent, monsters or other entities, objects and different types of terrain (such as walls, lava, water); ii) a statistics vector which indicates the agent's current $(x, y)$ location, hit points, time step $t$, and other features such as strength, dexterity and constitution, and iii) a textual message which gives different types of feedback about the environment ("you see a key of master thievery", "the wall is solid rock"). These are illustrated in Figure 4.

We now draw attention to some important subtleties regarding how existing methods implement count-based episodic bonuses—as we will see, these can have a large impact on performance. The works of [56, 25, 76], which use the MiniGrid suite [15] for evaluation, use a hash table whose keys are the full observations (in this case symbolic images) encountered by the agent. The occurrences of these observations are then counted to compute the episodic bonus. The baselines run in the MiniHack tasks [58], on the other hand, use a hash table whose keys are the $(x, y)$ positions of the agent, which are extracted from the observation and are then similarly counted to compute the episodic bonus.

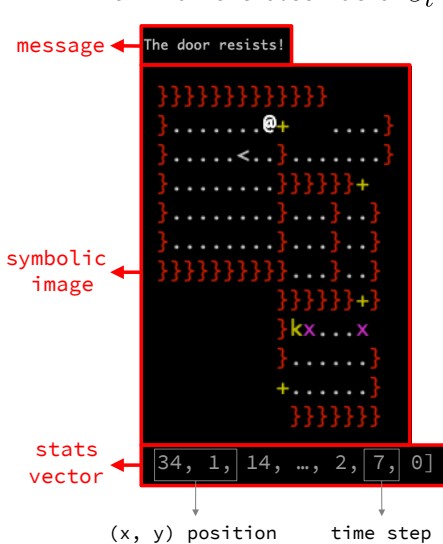

Figure 4: Observation for MiniHack

In order to understand the effect of these different input modalities on count-based episodic bonuses, we consider three variants of the NovelD algorithm, in addition to the original formulation. These methods all have a count-based episodic bonus of the form $\mathbb{I}[N_e(\phi(s_t)) = 1]$, with different choices of $\phi$ detailed below.

NOVELD: in the original version of the algorithm, $\phi$ is the identity. Here the input to the count-based episodic bonus is the full state, namely the concatenation of the symbol image, the message and the stats vector. This makes no assumptions about which part of the state is most useful.

NOVELD-POSITION: in this variant, $\phi$ extracts the $(x, y)$ position of the agent from the statistics vector, which reflects a strong inductive bias that the task is navigation-based.

NOVELD-IMAGE: in this variant, $\phi$ extracts the symbol image only. This reflects an inductive bias that the message and the stats vector are not useful since they are discarded.

NOVELD-MESSAGE: in this variant, $\phi$ extracts the message only. This reflects an inductive bias that the messages are important but the stats and symbolic images are not.

For E3B, we feed the full state to the algorithm and do not make any assumptions about which part is useful for the task at hand. Despite the lack of task-specific inductive biases, our method is still able to extract the relevant features for each task, thus outperforming the other exploration approaches (or matching their performance when prior knowledge of the task is used).

### 5.1.2 Results on MiniHack

Aggregate results for IMPALA, RND, RIDE, ICM, NovelD (with the three variants described above) and E3B over 16 sparse reward tasks from the MiniHack suite are shown in Figure 5. Out of 16 environments, 8 are based on the MiniGrid suite, but use the MiniHack interface and observation space (environment details are included in Appendix C.1.3). We used the performance metrics and bootstrapping protocol from [4] to compute confidence intervals, which are more informative than simple point estimates. We see that over all tasks (top row), E3B outperforms all other methods by a significant margin across all three performance metrics.

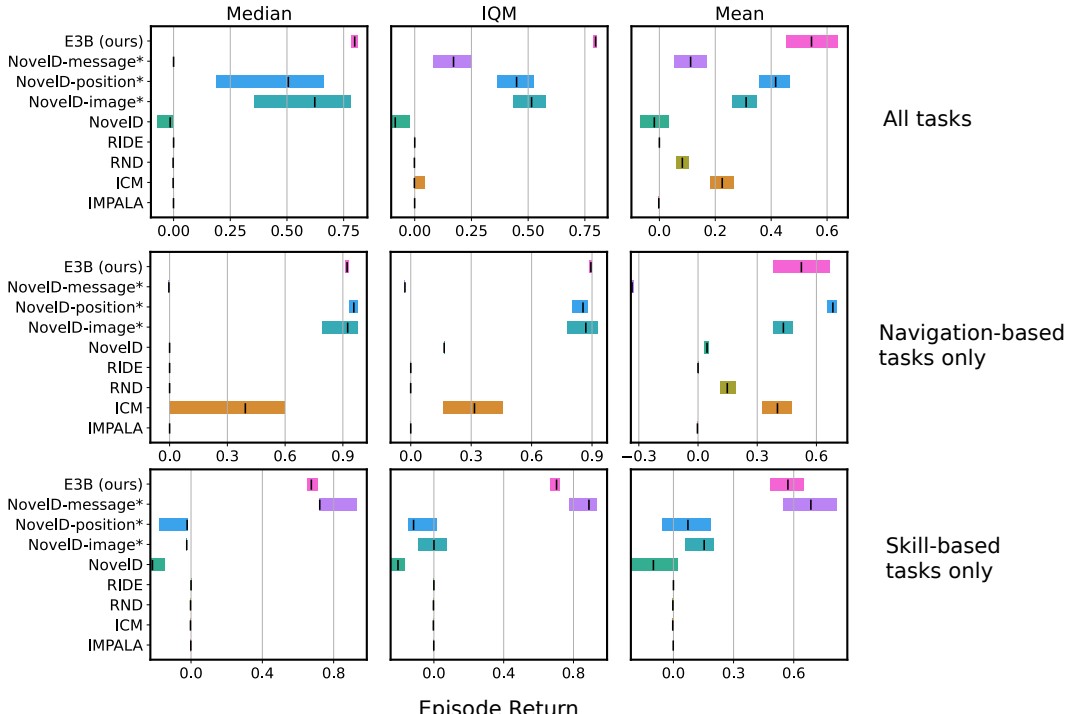

Figure 5: Aggregate results over 16 tasks from the MiniHack environment. Bars represent 95% confidence intervals computed using stratified bootstrapping with 5 random seeds. Methods marked with * use task-specific prior knowledge.

Standard NOVELD performs poorly due to the fact that the MiniHack observations contain a time counter in the statistics vector (see Figure 4), which makes each observation in the episode unique and hence renders the count-based episodic bonus meaningless. The three NovelD variants which use different features extracted from the state for the count-based bonus perform better. Figure 5 shows aggregate performance across a subset of 9 navigation-based tasks (middle row) and 7 skill-based tasks (bottom row) (see Appendix C.1.3 for the task breakdown). On the navigation-based tasks, NOVELD-POSITION has excellent performance, since visiting a large number of different $(x, y)$ locations is closely aligned with the true task reward. However, NOVELD-POSITION fails on all the skill-based tasks, resulting in poor performance overall. We observe an opposite trend for the NOVELD-MESSAGE variant, which performs very well on the skill-based tasks, but poorly on the navigation-based ones.

This highlights that although certain inductive biases can help for certain tasks, it is difficult to find one which performs well across all of them. Our method, E3B, performs well on both the navigation-based tasks and the skill-based tasks, resulting in superior performance overall. It is worth noting that E3B does not require any task-specific engineering: the exact same algorithm is run for all tasks, and it uses the unprocessed states. Results for individual tasks, along with more analysis and discussion, can be found in Appendix D.4.

### 5.1.3 Ablation Experiments

We next report results for ablation experiments measuring the effects of different algorithmic components of E3B, namely the feature encoding $\phi$ and the episodic nature of the bonus. Results are shown in Figure 6. E3B (random enc.) indicates E3B where $\phi$ is a randomly initialized network which is kept fixed throughout training. E3B (policy enc.) indicates E3B where the weights of $\phi$ are tied to those of the policy network (with the last layer producing action probabilities removed). E3B (non-episodic) indicates E3B where the

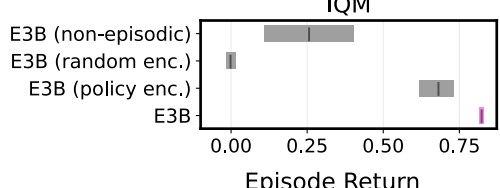

Figure 6: MiniHack ablations, bars represent 95% confidence intervals using stratified bootstrapping.

elliptical bonus is computed using observations across *all* timesteps in the lifetime of the agent, and not just the current episode. All three variants perform significantly worse than E3B, which uses an inverse dynamics model encoding for $\phi$ and an episodic bonus. This highlights that both the inverse dynamics model and episodic bonus are important for success.

## 5.2 Pixel-Based VizDoom

As our second evaluation testbed, we used the sparse reward, pixel-based VizDoom [37] environments used in prior work [52, 56]. Although these are singleton MDPs, they still constitute challenging exploration problems and probe whether our method scales to continuous high-dimensional pixel-based observations. Results comparing E3B to RIDE, ICM and IMPALA on three versions of the task are shown in Figure 7 (hyperparameters can be found in Appendix C.2). IMPALA succeeds on the dense reward task but fails on the two sparse reward ones. E3B is able to solve both versions of the sparse reward task, similar to RIDE and ICM.

We emphasize that these are *singleton* MDPs, where the environment does not change from one episode to the next. Therefore, it is unsurprising that ICM, which was designed for singleton MDPs, succeeds in this task. RIDE is also able to solve the task, consistent with results from prior work [56]. The fact that E3B also succeeds provides evidence of its robustness and its applicability to settings with pixel-based observations.

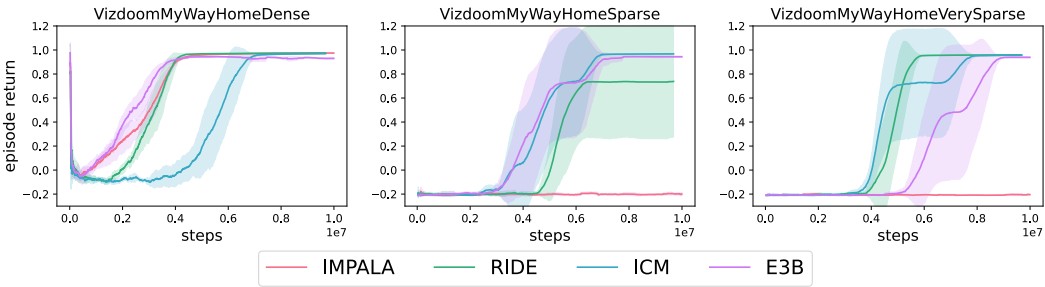

Figure 7: Results on pixel-based Vizdoom tasks with dense and sparse rewards. Results are averaged over 5 random seeds, shaded region indicates one standard deviation.

## 5.3 Reward-free Exploration on Habitat

As our third experimental setting, we investigate reward-free exploration in Habitat [59, 69]. Habitat is a platform for embodied AI research which provides an interface for agents to navigate and act in photorealistic simulations of real indoor environments. At each episode during training, the agent is initialized in a different environment, and it is tested on a set of held-out environments not used during training. These experiments are designed to evaluate exploration of realistic CMDPs with visually rich observations.

We use the HM3D dataset [57], which contains high-quality renditions of 1000 different indoor spaces. As our base RL algorithm we use DD-PPO [71] and train ICM, RND, NovelD and E3B agents using the intrinsic reward alone. We then evaluate each agent (as well as two random agents) on unseen test environments by measuring how much of each environment has been revealed by the agent's line of sight over the course of the episode. Full details on the experimental setup can be found in Appendix C.3.

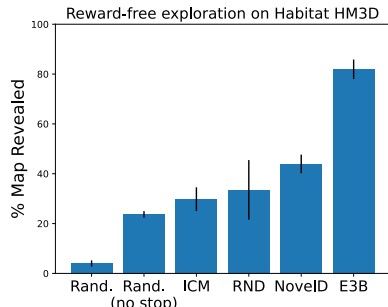

Figure 8: Reward-free exploration on Habitat. Error bars represent std. deviations over 3 seeds.

Quantitative results are shown in Figure 8. The E3B agent reveals significantly more of the test maps than any of the other agents. Trajectories for E3B, ICM, RND and NovelD on one of the test maps are shown in Figure 9, which illustrate how E3B explores a large portion of the space whereas the

others do not. These results provide evidence for E3B's scalability to high-dimensional pixel-based observations, and reinforce its broad applicability.

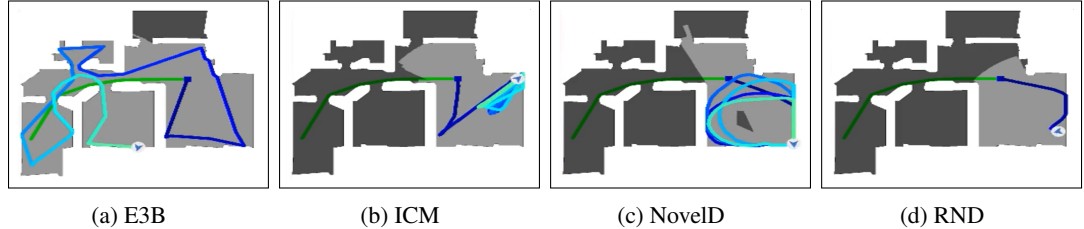

| (a) E3B | (b) ICM | (c) NovelD | (d) RND |

Figure 9: Trajectories of policies trained with different exploration algorithms, on a Habitat environment unseen during training. E3B reveals a larger portion of the map than other methods.

# 6   Related Work

**Exploration in RL.** Exploration remains a long-standing problem in RL. Common approaches include $\epsilon$-greedy [68], count-based exploration [66, 8, 48, 45, 70, 43], curiosity-based exploration [60, 64, 65, 11], and other types of intrinsic motivation [49, 50, 63, 1, 12]. These methods were largely designed for singleton MDPs, where the environment remains the same across episodes. As a result, they measure novelty over the agent's lifetime in a static environment rather than on an episodic basis under a distribution of diverse contexts, as necessary in CMDPs. Other intrinsic motivation methods have recently been developed for exploration in CMDPs [56, 73, 25, 76]. As our experiments show, they critically rely on episodic count-based bonuses, which E3B generalizes.

Another class of methods automatically generate curricula over variations of the CMDP to encourage efficient learning, effectively performing a form of curiosity-driven exploration in the context space. These include goal-conditioned [27, 26, 24, 55, 19, 14, 22] and goal-free variants [67, 54, 33, 21]. Unlike our method, these methods assume the ability to actively configure the environment context. In principle, E3B can be combined with these approaches, whereby E3B explores at an episodic level, while the automatic curriculum method explores at a context level, making the two complementary.

**Elliptical Bonuses.** The use of elliptical bonuses has a long history in the contextual bandit literature [6, 20, 42], which corresponds to the RL setting with a single time step. More recently, several works have begun to explore the use of elliptical bonuses in the context of multi-step RL. The PC-PG algorithm [2] considers MDPs with linear dynamics and uses an elliptical bonus based on a policy cover to explore in a provably efficient manner. The ACB algorithm [5] also uses an elliptical bonus, approximated using linear regressors trained on random noise, while FLAMBE [3] uses an elliptical bonus inside a learned dynamics model. All of these algorithms operate on singleton MDPs, whereas ours is designed for contextual MDPs and constructs elliptical bonuses at the episode level rather than across episodes. We also use different feature learning methods, namely inverse dynamics models, instead of the policy encoders, random networks or kernel methods used in the aforementioned works.

# 7   Conclusion

In this work, we identified a fundamental limitation of existing methods for exploration in CMDPs: their performance relies heavily on an episodic count-based term, which is not meaningful when each state is unique. This is a common scenario in realistic applications, and it is difficult to alleviate through feature engineering. To remedy this limitation, we introduce a new method, E3B, which extends episodic count-based methods to continuous state spaces using an elliptical episodic bonus, as well as an inverse dynamics model to automatically extract useful features from states. E3B achieves a new state-of-the-art on a wide range of complex tasks from the MiniHack suite, without the need for feature engineering. Our approach also scales to high-dimensional pixel-based environments, demonstrated by the fact that it matches top exploration methods on VizDoom and outperforms them in reward-free exploration on Habitat. Future research directions include experimenting with more advanced feature learning methods, and investigating ways to integrate within-episode and across-episode novelty bonuses.

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
