# OpenReview forum: "Exploration via Elliptical Episodic Bonuses"
_NeurIPS.cc/2022/Conference — NeurIPS 2022 Accept_

### Official Review · Reviewer_TeH7 · 2022-06-29

**Rating:** 8
**Confidence:** 3
**Soundness:** 3 good
**Presentation:** 3 good
**Contribution:** 3 good

**Summary:**

This work highlights an important weakness for existing exploration methods relying on intrinsic motivation in CMDPS, namely their over reliance on a count-based episodic term. Since this count-based term is computed on hand-engineered features, or on the entire observation, such methods underperform in noisy or more realistic environments. The authors propose to replace this term with an elliptical episodic bonus, computed on learned features extracted by inverse dynamic modeling. This simple modification significantly improves the performance of intrinsically motivated agents in complex environment which differ across episodes. This is shown through experiments on the MiniHack suite, and potential for scaling to image-based tasks is presented on VizDoom. The authors also report some interesting nuances in environments, and clarify the importance of heuristic choices in existing methods.

**Questions:**

**Main comments**
* The method's clarity can be improved. While the computation of the elliptical bonus is clear, its integration in the reward signal is not. This should be clearly discussed in the main body of the paper. Furthermore, to the best of my undestanding, E3B's reward only includes an episodic term, while RIDE, AGAC and NovelD also include a global term. While the lack of robustness in the episodic term is understandable, it is not clear why E3B also removes the global term.
* I would suggest author to ablate the method by replacing the elliptical terms with a hash-based count term [1], which can be computed over the representations learned through inverse modeling.
* The choice of MiniHack tasks is not well motivated. I am not fully informed on common evaluation protocols on MiniHack, but to my understanding the authors performed a selection of tasks. The reasoning behind this selection should be motivated, particularly in relation to prior works.
* Figure 6 shows how RIDE does not achieve any reward signal in any of the tasks considered. Can the authors provide an intuitive explaination for this result?

**Minor comments**
* line 50: Prior works on elliptical bonuses should be cited before the Related Works section on the last page. I would suggest to add them here, in order to clarify that this idea is not novel.
* lines 83-85: I would recommend adding references to NovelD and AGAC to better assist the reader.
* Figure 1: What is the behavior of each agent when disabling the count-based episodic bonus? Is their explorative behavior less extended, or do they fail to move at all?
* line 123: This experiment is only reported for NovelD, but the reasoning for this choice is only provided on line 243. I would recommend to move this explanation here.
* line 176: Why is $n$ not fixed across all experiments? How is its value selected?

**References**

[1] Haoran et al. "#Exploration: A Study of Count-Based Exploration for Deep Reinforcement Learning.", 2017

**Limitations:**

The authors do not overstate their contribution, and the limitations of their work are properly presented.

**Strengths And Weaknesses:**

**Strengths**
* The claims made through the paper are modest, but well argued and sufficiently supported by experimental results.
* The paper is well written and easy to follow. The motivating weakness of existing methods is outlined well, and the idea of elliptical bonuses is clearly presented, and motivated from different point of views.
* The experimental section is convincing: the metrics reported follow a statistically principled evaluation approach, baselines are satisfactory and their behavior is well explained.

**Weaknesses**
* While the motivation and the intuition of the method are clear, the main body of the paper does not explicitly describe the algorithm itself, which is relegated to the Appendix. As a result, this part of the paper is lacking in clarity.
* The method is not particularly original, as it can be seen as an incremental generalization of count-based exploration bonuses to continuous settings.

---

> ### Author Response · Authors · 2022-08-02
> **Response**
>
> $~$
>
> Thank you for your insightful review, your suggestions to improve the paper, and your support for our work. We are glad you thought the experimental section was convincing and the claims well-argued. We have uploaded a revision based on your suggestions and address the comments below.
>
> - [_While the motivation and the intuition of the method are clear,..._] We would have liked to include the algorithm in the main text itself, but were limited by space constraints. However, since there is an additional page allowed for the camera-ready, we will add this to the main text if the paper is accepted.
> - [_The method is not particularly original, as it can be seen…_] We agree that the idea of elliptical bonuses has been around for a long time and is not new, and we clarified this by adding references in the introduction as suggested. We believe the key contributions of our work are: i) highlighting the importance of the count-based **episodic** bonus in prior approaches for CMDPs, showing why it fails in more realistic scenarios, and ii) fixing this problem by proposing the elliptical **episodic** bonus. To our knowledge, using the elliptical bonus at the episode level has not been done before (prior approaches compute the bonus using all previous episodes), and this is essential for good performance in CMDPs, as evidenced by our ablation in Section 5.1.3 where using the non-episodic elliptical bonus does much worse. Furthermore, existing works using elliptical bonuses in deep RL settings with singleton MDPs (PC-PG, ACB) use different feature extraction methods which we show do not work in our setting.
>
> $~$
>
> ### Main questions:
>
> - [_The method’s clarity can be improved…_] We have clarified how the bonus is integrated with the extrinsic reward function in Section 4.1. Concerning the global term, in initial experiments we also included NovelD’s global term, but this performed similarly to the elliptical episodic bonus alone, so we removed it to have a simpler algorithm.
> - [_I would suggest the authors to ablate…_] That is a very interesting suggestion. We didn’t have the time to do this in the revision period, but will look into it for the camera ready.
> - [_The choice of MiniHack tasks is not well motivated…_] We have clarified this in the paper. As our starting point, we chose the MiniHack tasks from the original MiniHack paper which were not solvable by the methods evaluated in that paper (such as MultiRoom-N4-Locked, MultiRoom-N4-Lava). We then made some of these harder, by adding rooms (i.e. making N6, N10 versions of the tasks). All of the skill-based tasks are taken from the official github repository. We also added several other navigation-based tasks from the github repo: Labyrinth-Small and Labyrinth-Big, and the LavaCrossS\*N\* tasks. We also added some new task variants such as MultiRoom-OpenDoor. These are interesting because they highlight the brittle nature of some of the count-based heuristics such as NovelD-message. Indeed, NovelD-message is able to solve the closed door variants, because opening the door causes a message to appear, but it fails to solve the open door variants because no messages appear which provide a novelty bonus. This phenomenon is discussed in Appendix D.4, and more details about the environments are given in Appendix C.1.3. To our knowledge, this work performs the most comprehensive evaluation of MiniHack tasks to date.
> - [_Figure 6 shows how RIDE does not achieve any reward signal…_] First, the RIDE results in Figure 6 use the standard episodic count-based bonus (without heuristics like the ones we used for NovelD, i.e. extracting (x,y) positions or messages), therefore due to the time counter feature in MiniHack this bonus will be constant. Based on our results in Section 3, this is enough to make the algorithm fail. We also tried some limited experiments using the same position-based heuristic as NovelD, but weren’t able to get good results either. Note that the original MiniHack paper also reports that RIDE with a count-based bonus based on (x,y) positions did not give any improvement over IMPALA.
>
> $~$
>
> ### Minor comments:
> - [_line 50: Prior works…_] done
> - [_lines 83-85: …adding references to NovelD and AGAC…_] done
> - [_Figure 1: What is the behavior of each agent…_] We will check this for the final version of the paper.
> - [_line 123: this experiment is only reported for NovelD…_] done
> - [_line 176: Why is $n$ not fixed across experiments?..._] We set the $\phi$ encoder to have the same architecture as the policy network, and thus set $n$ to be equal to the number of hidden units at the last layer. This was 1024 for MiniHack and 256 for Vizdoom.

---

> > ### Comment · Reviewer_TeH7 · 2022-08-05
> > **Reviewer Response**
> >
> > I would like to thank the authors for addressing the concerns that were raised, and actively improving the paper during the rebuttal phase. Considering the changes to Section 5.2, and the detailed clarifications, I have updated my score.

---

### Official Review · Reviewer_yKuH · 2022-07-05

**Rating:** 7
**Confidence:** 3
**Soundness:** 4 excellent
**Presentation:** 4 excellent
**Contribution:** 3 good

**Summary:**

The paper gives evidence of the importance of the pseudo count quantity used in existing exploration algorithms in RL, and proposes a new algorithm to extend the pseudo count idea to continuous state spaces by also proposing a method to learn a feature encoder. They give this evidence through various empirical experiments.

Thanks to the authors for putting in the effort in doing this work!

**Questions:**

Questions/Suggestions:
- How are differences between contexts defined? I see that there is a distribution over contexts but how extreme can differences between two contexts be? It doesn't appear to be the case here, but generally does the reward function change between contexts too?
How many samples are actually needed for fitting in the ellipse?
- The paper also notes that “any feature learning method” could be used, but it proposes to use the inverse dynamics model. I am curious how off-the-shelf feature extractors perform with the elliptical exploration algorithm? It seems like this problem is an independent problem itself, and it would make sense to leverage existing works instead of re-inventing something new.
- How sensitive are results to the regularization term in Equation (1)? In my experience, I have found this term to be sensitive.
- I know the goal isn't explicitly lifelong learning, but I am curious what the authors think about getting E3B (non-episodic) from Figure 7 to work in that setting? In a lifelong setting where the agent is constantly interacting with no clear start and end to different contexts, I wonder how the proposed method would work then and how it compares to existing methods?
- So we can see a clear performance improvement in Figure 6. But I am curious why ICM performs similarly to E3B on the high-dimensional Vizdoom domain, especially given that the motivation was current methods don’t fare well in high dimensional/continuous state spaces?
- How will the count-based methods in Table 1 improve or change if you use the proposed feature encoder (inverse dynamics model) as input into these existing exploration methods instead of using the proposed elliptical count-based algorithm?


**Limitations:**

Yes, they have.

**Strengths And Weaknesses:**

Strengths:
- I appreciate works that take a hard look at existing methods in the manner that is done in this paper and try to understand the fundamental flaws.
- I like table 1 of distilling the exploration methods to their essence, and also I like figure 3.
- The time-counter example of how states can appear unique is a nice one for intuition.
- The paper is generally well-written and easy to follow.

Weaknesses:
- I view the paper as trying to solve two independent problems, but the paper makes it seem they are intrinsically tied together. The two problems are: 1) learning a feature encoder and 2) exploration. While the paper does include an ablation study showing how important both these parts are, it would be nice to get a better understanding of these two independent components. For example, how does using the learned encoder improve other existing count-based methods, or how does using off-the-shelf feature encoders affect the performance of the elliptical exploration algorithm?

---

> ### Author Response · Authors · 2022-08-02
> **Response 2/2**
>
> $~$
>
> ### Questions:
>
> 1. [_How are differences between contexts defined?..._] For procedurally-generated CMDPs such as MiniHack/MiniGrid, the context can be thought of as the random seed determining the generation process. How much difference this induces between two different environments depends on the task. For example, for the MiniHack-MultiRoom tasks, different random seeds cause very different map layouts and number/type of monsters. For others, the spatial layout is the same but different seeds cause there to be different objects which the agent must use. In all our settings, the reward function is defined the same way. For Habitat, different contexts correspond to different simulated houses, which can differ a lot in their spatial layout and visual characteristics.  Concerning the ellipse, it is fit using all the previous samples in the current episode. So after $t$ steps in the current episode, we use $t-1$ samples to fit the ellipse and compute the exploration bonus.
> 2. [_The paper also notes that “any feature learning method” could be used, but…_] If there is a natural off-the-shelf feature extractor, then indeed this could be used in place of the feature extractor learned with the inverse dynamics model. Currently, we are not aware of any pretrained feature extractors for NetHack/MiniHack, so we chose the inverse dynamics model. But we agree that it is interesting to compare the performance of different feature extractors. In Figure 7 we compare to two other ones: a simple randomly initialized network (even though this is simple, some other works have shown that this can work surprisingly well in some cases, e.g. [11, 2]), and the policy trunk (which has been used in [5]). Out of these, the feature extractor learned with the inverse dynamics model worked the best.
> 3. [_How sensitive are results to the regularization term in Equation (1)?..._] We ran experiments with different values of the $\lambda$ parameter and added results in Appendix D.5. Compared to our default value of $\lambda=0.1$, there is no statistically significant drop for $\lambda=0.01$, but $\lambda=0.001$ and $\lambda=1.0$ are a bit worse. This shows that the $\lambda$ parameter is robust across an order of magnitude.
> 4. [_I know the goal isn't explicitly lifelong learning, but I am curious what the authors think…_] That’s an interesting question! One approach could be to define the covariance matrix as follows: $C_t = \sum_{i=1}^t \alpha^{t-i}\phi(s_i)\phi(s_i)^\top + \lambda I$ for some $\alpha<1$ (say $\alpha=0.99$ or $0.95$). This way, the contribution of samples in the past to the covariance matrix decays smoothly with time. This can be implemented by updating $C_t = \phi(s_t)\phi(s_t)^\top + \alpha C_{t-1} + \lambda I$. However, this gets a bit tricky because we can no longer use rank-1 updates to compute the inverse. An alternative could be to constrain the $C_t$ matrices to be diagonal, in which case inversion is easy and we don’t need rank-1 updates anymore. In preliminary experiments we found that the diagonal approximation still worked well, but this would need to be tested more thoroughly.
> 5. [_So we can see a clear performance improvement in Figure 6. But I am curious why…_] We believe the reason is that the Vizdoom tasks are singleton MDPs - i.e. the environment is the same every episode. This is in contrast to contextual MDPs (like MiniGrid/MiniHack/Habitat), where the environment changes every episode. ICM was originally designed for singleton MDPs (in fact, Vizdoom was one of the tasks used in the original ICM paper), so it makes sense that it works well here. What we mean in Section 3 is that the count-based episodic bonus component doesn’t fare well in large/continuous state space. This seems an important driver of performance for CMDPs, but it’s not clear that it’s important for singleton MDPs. For singleton MDPs, the other terms in the bonus may be sufficient for exploration. **Note also that in our new results with Habitat (which is a CDMP), ICM performs much worse than E3B**.
> 6. [_How will the count-based methods in Table 1 improve or change…_] If every state is distinct, even if we encode them using the feature extractor learned with the inverse dynamics model before feeding them to the count-based term, it is unlikely to work because the encoder would have to map two distinct states to exactly the same embedding for the count-based bonus to treat them as the same. Even if the distance in embedding space is tiny, the count-based bonus will treat them as separate. However, one could replace the count-based bonus in any of the methods by the elliptical bonus. This was our original approach, but we later found that using the elliptical bonus alone gave equally good performance so we stuck with that to make the algorithm simpler. However, there might be settings in which combining a cross-episode bonus with an episodic bonus would be helpful, which we hope to investigate in future work.

---

> > ### Comment · Reviewer_yKuH · 2022-08-05
> > **response to authors**
> >
> > Thanks to the authors for addressing the concerns and updating the paper. I will stick with the current score. This is a nice paper.

---

> ### Author Response · Authors · 2022-08-02
> **Response 1/2**
>
> $~$
>
> Thank you for your detailed and insightful review, your suggestions to improve the paper, and your support for our work. We are glad you enjoyed the paper’s look at the limitations of existing methods and ways to fix them. We answer your questions in detail below.
>
> - [_I view the paper as trying to solve two different problems_] Please see our answers to questions 2 and 6 below.

---

### Official Review · Reviewer_rjcQ · 2022-07-06

**Rating:** 4
**Confidence:** 4
**Soundness:** 2 fair
**Presentation:** 4 excellent
**Contribution:** 2 fair

**Summary:**

The paper proposes a method that alternates between learning the feature mapping via the inverse dynamics model and using the learned feature to construct the elliptical bonus for exploration. The paper also analyzes several heuristic exploration bonus method by removing the count-based component in them and shows that they fail under such modification. The paper claims that the proposed algorithm contributes the most to the contextual markov decision processes (CMDP) setting, and shows its empirical competitiveness in the MiniHack benchmark.

**Questions:**

1. In figure 1, why are the baselines not performing well even with their original constructions (with counts)?

2. The motivation against using counting based method seems not very fair? "if each state is unique" (line 102) almost implies that the state space is continuous, then one obviously should never use naive count-based bonus, since count-based bonus only works for tabular MDP.

3. In the feature learning part, it is known that the inverse dynamics model is valid only if it is conditioned on some conditional distributions of actions (for example, policies). In the proposed algorithm, is the conditioned policy being a mixture of all previous policies? Would it providing conflicting information, for example, at some state policy a goes left and policy b goes right? Since the algorithm aims to perform exploration.

4. In result shown in Fig.6, is there any hypothesis why RND completely fails on all tasks?

5. The ablation shown in section 5.1.3 is not using very strong baselines. For example, in [1,2,4], those method are using RFF kernels and random networks like RND as feature mappings, which show convincing empirical performance.

6. In Fig.8, is there any possible explanation for why E3B is worse than the baselines in some task?

**Limitations:**

1. As mentioned in the previous section, resetting the covariance matrix seems a limitation of the proposed work.

2. The results shown in Section 5.2 shows that the proposed algorithm achieves very limited improvement or even no improvement than previous algorithms on regular MDPs with large state space.

**Strengths And Weaknesses:**

## Strength

1. The proposed method, E3B, when evaluated in the MiniHack benchmark, which is a reasonable benchmark for CMDP, shows better performance than the previous empirical methods with heuristic exploration bonuses. The evaluation also contains some variants of the baselines to provide more insights into the results.

2. The paper is easy to follow with several informative visualizations.

## Weakness

1. It is very surprising to see that the proposed algorithm, which resets the empirical covariance matrix at the beginning of the episode (according to Algorithm.1), can actually work. First, this should not work in theory, where your covariance matrix $C$ is constructed using the policy cover defined on all previous iterations. Intuitively, it is also not very clear why this can work effectively. Resetting $C$ at the beginning of the episode means the bonus construction has no memory of what states have already been visited before. Combined with that the algorithm performs policy optimization based on reward + bonus, consider the setting if we have sparse reward and short horizon: for example, if one wants to escape a maze in small time budget, the algorithm could first visit some dead end, but receives high bonuses due to resetting $C$, and update to visit the dead end with higher probability after the PG update, and visit the dead end again but still receives high bonus, and thus eventually stuck?

2. The contribution of this work does not seem obvious, either. Using elliptical bonus in Deep RL or with neural network function approximation has already studied by some previous work. For example, PC-PG [1] (which this paper also mentions) actually uses PPO+elliptical bonus in their experiment. [2] even has a specific Deep RL version that uses deep RL + elliptical bonus and is evaluated in the deep RL benchmarks. If we also consider the representation learning part, [3] provides both theoretical and empirical results with neural network function approximation. In addition, [4] also justifies that a variant of [2] is actually performing feature learning in a noisy system. There seems no comparison against any of these works in this paper, given that the approaches are similar and the above baselines actually also show strong performance in practice.

### references

[1] Agarwal, Alekh, et al. "Pc-pg: Policy cover directed exploration for provable policy gradient learning." Advances in neural information processing systems 33 (2020): 13399-13412.

[2] Song, Yuda, and Wen Sun. "Pc-mlp: Model-based reinforcement learning with policy cover guided exploration." International Conference on Machine Learning. PMLR, 2021.

[3] Xu, Pan, et al. "Neural contextual bandits with deep representation and shallow exploration." arXiv preprint arXiv:2012.01780 (2020).

[4] Ren, Tongzheng, et al. "A free lunch from the noise: Provable and practical exploration for representation learning." arXiv preprint arXiv:2111.11485 (2021).

---

> ### Author Response · Authors · 2022-08-02
> **Response 2/2**
>
> $~$
> ### Contribution compared to PC-PG and PC-MLP
>
> As noted above, the focus of this work is on contextual MDPs, where the environment changes at each episode. Although PC-PG and PC-MLP also use elliptical bonuses, a major difference is that they are designed for singleton MDPs. The assumption that the environment is the same at each episode is central to both algorithms: they both operate by iteratively growing a policy cover that progressively covers the state space of the MDP, and define the reward bonus to be high outside the covered region using the covariance matrix of previous policies computed over previous episodes. However, to go back to the maze example above, if the maze changes from one episode to the next, the covariance matrix of features computed over one episode no longer makes sense in the context of a different episode, because they each correspond to different mazes. Therefore, although they both use an elliptical bonus, E3B and PC-PG/PC-MLP are designed for very different settings. E3B was designed for the CMDP setting which is a more general and challenging one, whereas PC-PG and PC-MLP were designed for the simpler MDP setting. As our experiments show, other exploration methods which are very effective in singleton MDPs such as RND or ICM, are significantly worse than E3B when applied in CMDPs.
>
>
> $~$
> ### Questions:
>
> - [_In figure 1, why are the baselines not performing well…_] For all experiments in Figure 1, we used the official code provided by the authors of the algorithms with the recommended hyperparameters. We believe the fact that the methods do not completely solve the task is due to the difficulty of the task. In Figure 12 of Section D.1 in the Appendix, we provide a similar ablation on two other MiniGrid environments where some of the methods perform better (for example, NovelD gets close to 1 return). Our main point here is that **if the episodic count-based term is removed, these baselines algorithms fail to learn at all**.
> - [_The motivation against using counting-based method seems not very fair…_] We agree that count-based bonuses are ill-suited for large/continuous state spaces. However, existing published state-of-the-art works for CDMPs (RIDE, AGAC and NovelD) all use a count-based episodic bonus, and we show in Section 3 that they all perform poorly if it is removed. Our point is that using a measure of episodic novelty is essential for good performance in CDMPs, but the existing way of doing this (using a count-based episodic bonus) does not scale to large/continuous state spaces. This is why we introduce the episodic elliptical bonus, which also measures episodic novelty, but does scale to large/continuous state spaces.
> - [_In the feature learning part, it is known that the inverse dynamics model is valid only if it is conditioned…_] The inverse dynamics model and feature extractor are trained using the on-policy data generated by the policy. However, note that it maps (s, s’) pairs to actions. Therefore, if the policy takes action $a_1$ at state $s$ the first time, and $a_2$ at $s$ the second time, these will likely generate different next states $s’$ which will allow the inverse dynamics model to disambiguate between $a_1$ and $a_2$. If we have two different actions which induce the same distribution over next states, then this may not be a well-defined problem, but this is probably rare in practice and may not affect the features learned by $\phi$.
> - [_In result shown in Fig.6, is there any hypothesis why RND completely fails..._] We hypothesize that this is because RND constructs its bonus based on all previous episodes, which is often not appropriate for CMDPs since the environment changes each episode (see our maze example in the section above about resetting covariance matrices).
> - [_The ablation shown in section 5.1.3 is not using very strong baselines…_] We did run E3B using a random network encoder, this is denoted E3B (random enc.) in Figure 7. Please let us know if this is not clear. It is not possible to directly run a baseline with RFF kernels, because the MiniHack inputs are mostly symbolic.
> - [_In Fig.8, is there any possible explanation for why E3B is worse than the baselines in some task?_] E3B matches the final performance of all baselines. It requires slightly more samples for the last task, but we did not tune it much and it’s possible that it would converge faster with some more tuning. In general, we believe it’s hard to draw conclusions from convergence speed unless the difference between two algorithms is really big, because one can always tune methods more to try and make them converge faster.
>
> $~$
> ### Summary
>
> We hope the clarifications above and the new experimental results on a photorealistic 3D environment are sufficient for you to reconsider your assessment of the paper. Please let us know if you have any outstanding concerns that stand between us and a strong recommendation for acceptance.

---

> > ### Comment · Reviewer_rjcQ · 2022-08-07
> > **Responses to the authors**
> >
> > First I would like to thank the authors for their detailed response and also apologize for my late reply and certain overlooks in the original review (for example, the random network baselines). I would address some major issues in this followup:
> >
> > 1. First I want to further discuss the contextual mdp setup. I am aware of the setup in my original review but I would provide some additional clarification since it causes some confusion during the discussion. I would argue that we should consider CMDP as a broader class of MDP, because for example, we could consider a singleton mdp as a contextual mdp whose context distribution only has one non-zero point-mass. I appreciate the authors' acknowledgment that my previous maze counter-example would make sense in the singleton mdp case, but we can easily extend it to cmdp case because we can assume everytime the dynamics are nearly identical each iteration (and deterministic), and the goal is sampled in some state that the policy never visited before, then I believe the counterexample should still hold.
> >
> > 2. However, the above argument could be treated as an argument against episodic bonus overall. Here I thank the reviewer for the pointer to section 3 which shows that the episodic bonus is indeed helpful even for other algorithms. Thus it indeed seems like episodic bonus is helpful for practical cmdp problems that enjoy certain structures.
> >
> > 3. I appreciate the authors for acknowledging that counting-based bonuses are ill-suited for large/continuous state spaces. I understand the motivation of section 3 for showing the failure of count-based methods, but I still can not appreciate the way the current motivation is formulated because the ineffectiveness of counting based bonuses for large/continuous state spaces is just too obvious. In addition, if we want to talk about small state/action space, I would also like to point out that **elliptical potential bonuses reduce to count-based bonuses in tabular case.**
> >
> > 4. For the positive side, I am very impressed by the new experiments because the habitat benchmark seems a challenging environment.
> >
> > 5. For a side note, one thing we have been ignoring is that the algorithm is *learning* the feature and the bonuses are constructed by the learning feature. Although again previous algorithms have been studying learning feature + elliptical bonuses, the practical advantage of learning such features on the fly seems still unclear. For example, if we use *ground-truth* feature + elliptical episodic bonuses in the above maze counterexample, it does not seem to work. However, using a shifting feature (even though it is incorrect!) seem to break the deadlock (even though it's not clear to me if it is in a good way or a bad way).
> >
> > In summary, I would acknowledge that my major critique still remains, but mostly to the episodic bonuses. However, the most major difference between this work and previous (learning feature + elliptical bonuses) methods is the reset of the covariance matrix (thus episodic bonus), which as I argued above, remains not obvious why it would work without nice context generation property. While I highly appreciate the new experiment result, I would still like to keep my original evaluations.

---

> > > ### Author Response · Authors · 2022-08-08
> > > **Response 2/2**
> > >
> > >
> > >
> > > References:
> > >
> > > [1] Kirk et al. "A survey of generalisation in deep reinforcement learning"
> > >
> > > [2] Juliani et al. "Obstacle tower: A generalization challenge in vision, control, and planning."
> > >
> > > [3] Cobbe et al. "Leveraging procedural generation to benchmark reinforcement learning."
> > >
> > > [4] Küttler et al. "The NetHack learning environment"
> > >
> > > [5] Samvelyan et al. "MiniHack the planet: A sandbox for open-ended reinforcement learning research"
> > >
> > > [6] Guss et al. “The minerl competition on sample efficient reinforcement learning using human priors”
> > >
> > > [7] Chevalier-Boisvert et al. “Babyai: A platform to study the sample efficiency of grounded language learning”

---

> > > > ### Comment · Reviewer_rjcQ · 2022-08-09
> > > > **Response to the authors**
> > > >
> > > > I appreciate the authors for their reply. Here I would summarize my arguments with some updates:
> > > >
> > > > 1. We agree that the elliptical bonus is suitable for continuous state space (let's put the issue of whether it is episodic aside for now), and naively using count-based bonus obviously does not make sense. I would restate that the elliptical bonus is widely studied both in theory and in practice, so I can not agree this is actually one of the contributions of the paper.
> > > >
> > > > 2. Regarding the proposed episodic variant of the elliptical bonus, I would argue the example I mentioned above is actually just a very simple and general example without any specific adversarial construction, so I don't agree it is not representative and can be simply ignored.
> > > >
> > > > 3. Regarding the author's claim on the importance of cmdp, I actually think this is the major source of our disagreement. The authors agree that in order for the episodic bonus to work, the problem has to enjoy a certain favorable structure. I think this is why previous papers such as [1] claim that their episodic reward is for "procedurally generated environments" instead of claiming to solve cmdp, and there seems no addressing of this limitation in the revised paper.
> > > >
> > > > 4. I would like to bring up an issue of the ablation study that the authors mention. In the ablation study, the authors compare the episodic version and non-episodic version of their constructed bonus. However, according to the efficient computation of the covariance matrix that the authors propose in the paper, if they compute the non-episodic bonus in this way, the covariance matrix is aggravated by all previously learned features, which does not make sense anymore because one should use the latest feature and use all previous samples to recompute the covariance matrix from scratch. Please correct me if I understand the construction wrong.
> > > >
> > > > In summary, I agree, as I mentioned in the previous threads, that the paper shows good empirical results. However, again, the novelty and contribution of the proposed method, compared with previous (learning feature + elliptical bonuses) methods, seems only the episodic version of the bonus for cmdp. The contribution seems limited to me, and I am not convinced episodic bonus is the solution to cmdp. Other factors (such as the changing feature as I brought up earlier) may also result in a boost in performance and I believe a more careful empirical analysis is required. I argue that regardless of the nature of the paper, one algorithm should match its claim at least on an intuition level but I am not convinced in the current form of the paper. Finally, I acknowledge that the wording of my current rating is overly harsh: thus I increased my rating by 1 but I can not assign a score higher than this because that would contradict my overall evaluation of the paper.
> > > >
> > > > [1] Ride: Rewarding impact-driven exploration for procedurally-generated environments.

---

> > > > > ### Author Response · Authors · 2022-08-09
> > > > > **Response**
> > > > >
> > > > > Thank you for reading our response. Although we still respectfully disagree with the current rating, we do appreciate you raising your score. We address the concerns you bring up below:
> > > > >
> > > > > $~$
> > > > >
> > > > > 1. Although you say “naively using count-based bonus obviously does not make sense”, we would like to point out that the three best previously published methods for CMDPs that we consider (RIDE, AGAC and NovelD) _all_ use an episodic count-based bonus. Also note that in the papers introducing these methods, the episodic count-based bonus is mentioned as a heuristic rather than the central algorithmic contribution. Our point is this: on one hand, the importance of the episodic count-based bonus has been underestimated by the community, while at the same time, the reason its limitations haven’t been more apparent is because these algorithms have been evaluated on simple MiniGrid environments where there is a small number of states per context (and hence, the count-based bonus works). If we run these algorithms on more complex environments with a large number of states per context, they fail because the count-based bonus is no longer meaningful. **Therefore, we i) point out a major limitation of existing methods, and ii) propose a solution. We believe these are important contributions to the community, even more so because our fix is simple and easy to integrate into other algorithms.**
> > > > >
> > > > > $~$
> > > > >
> > > > > 2. First, we would like to note that the example you describe (a maze with multiple dead end hallways) is similar in nature to the Bidirectional Diabolical Combination Lock in the PC-PG paper, which requires not just an appropriate bonus construction, but also more sophisticated mechanisms such as a policy cover to solve reliably (note RND, which uses a non-episodic bonus, also fails on that task due to using a single policy). As we mentioned above, it’s not obvious how to use a policy cover in a CMDP where the same environment is never seen more than once. **Second, this is just one example environment, whereas our experiments use over 20 environments, including ones based on real-world scenes. While we do not claim our algorithm works in _every_ CMDP, it works on many of the ones used in empirical research, and serves as an important foundation for future refinements.**
> > > > >
> > > > > $~$
> > > > >
> > > > > 3. We unfortunately do not follow the reviewer’s reasoning, and do not understand the limitation they are referring to. **Procedurally-generated environments are a type of CMDP, and are the most common type used in empirical research. We also evaluate our algorithm on Habitat, which is a type of CMDP which is not procedurally-generated.** Finally, it is clear that there must be some shared structure among the different environments instantiated by the CMDP - otherwise there is nothing to generalize from. This holds true even in the dense reward case where exploration is not necessary, since the policy must generalize between different environments corresponding to different contexts.
> > > > >
> > > > > $~$
> > > > >
> > > > > 4. **Although what you describe would be ideal, unfortunately this is not computationally feasible**, since it would require storing all the agent’s experience (up to 50 million samples total), and would require running all of these through the feature encoder every time it is updated (currently, we update it approximately every 80 steps). Even using the naive matrix inversion rather than the efficient rank-1 updates causes a slowdown of 3x, which is not practical (currently, training one agent on one environment takes over 20 hrs on a modern machine with one GPU and 40 CPU cores).
> > > > >
> > > > > $~$
> > > > >
> > > > > We thank you again for engaging with our paper. At this stage, we are unsure of what actionable steps we can take to update our paper so that you will consider increasing your score, but we are open to suggestions and happy to run additional specific experiments (within reason) for the final version. Nevertheless, we hope the above clarifications have fully addressed your concerns and you will consider recommending acceptance.

---

> > > ### Author Response · Authors · 2022-08-08
> > > **Response 1/2**
> > >
> > > Thank you for your engagement in the discussion and your help in improving the clarity of our paper. That said, we are honestly quite puzzled by your assessment of the paper at this point. To summarize, it seems we agree on the following points:
> > >
> > > - Empirical results of our method for the challenging pixel-based photorealistic CMDP Habitat are impressive
> > > - CMDPs are an important class of RL problems [1] with more and more community benchmarks building upon the framework (e.g., [2]-[8] to name just a few)
> > > - Episodic bonuses are helpful for practical CMDP problems that enjoy certain structures
> > > - Naively counting states (using the identity function) is ill-suited for constructing episodic bonuses in large/continuous state spaces (note that this is exactly what our method is addressing)
> > >
> > > In our view, we have clearly demonstrated that our method, building on elliptical bonuses, overcomes the problem of naively counting states for episodic bonuses, that it empirically beats state-of-the-art exploration methods on a wide range of established CMDP benchmark environments, and that it even works for very challenging pixel-based photorealistic 3D environments (Habitat).
> > >
> > > With all respect, if you agree with the statements above (which as far as we understand, you do), then we fail to see how a "3: Reject: For instance, a paper with technical flaws, weak evaluation, inadequate reproducibility and incompletely addressed ethical considerations." is justified.
> > >
> > > Therefore, we would appreciate more clarification as to why you still think resetting the covariance matrix is a concern. **We have explicitly run a version of our algorithm where the covariance matrix is _not_ reset, and it performs much worse.** This is in Figure 6, called "E3B (non-episodic)". Furthermore, in our response above we have provided intuition for why resetting the covariance seems appropriate for CMDPs where the environment changes each episode (see paragraph 3 in our section on resetting the covariance).
> > >
> > > **Note that our paper is not a theory paper but an empirical one**. We do not make any (theoretical) claims that our approach will work in _every_ possible CMDP, but rather demonstrate empirically that our approach outperforms other SOTA methods on many established challenging CMDP benchmarks, with more than 23 tasks in total. Thus, we believe our thorough empirical evaluation of this approach should be enough to warrant acceptance.
> > > We acknowledge that it would be interesting to understand from a theoretical standpoint _why_ our method works when it does, and also characterize the cases when it does not work, but this is outside the scope of the current paper. In fact, it often is the case that an algorithm’s effectiveness is first demonstrated empirically and later on a theory is developed to better understand why. We hope our paper will inspire other researchers to further investigate this. However, we do not think that the existence of one possible counterexample invalidates our entire approach and is cause for rejection.
> > >
> > > **We would like to emphasize the thorough nature of our empirical results**: our algorithm outperforms 8 competitive baselines, including ones _specifically designed for CMDPs, as well as new variants of existing baselines (such as NovelD-message, NovelD-pos) which we have designed and which improve upon the original baselines themselves_, on 22 diverse CMDP tasks from MiniHack, as well as in a 3D photorealistic setting from Habitat, which are based on _real world_ indoor settings.
> > >
> > > Finally, we would like to point out that we are addressing a considerably more general and challenging setting than previous work that only considers singleton MDPs. While our current algorithm may not perform as well as methods specifically tailored for singleton MDPs (like PC-PG/PC-MLP) on certain pathological MDP problems, methods like PC-PG are also expected to fail on CMDPs where the environment changes each episode, as noted in our response above (because the covariance of features computed over one episode no longer makes sense in the context of another episode, since the environment has changed). Thus, we are not aware of any other approach that performs as well as ours in practice on such a wide range of challenging CMDP benchmarks and we believe this in itself is a valuable contribution that other researchers could build on.
> > >
> > > We thank you for your review and consideration. We made a significant effort to address your concerns, and would greatly appreciate it if you would consider raising your score in light of our response. Please let us know if you have any final questions that we could further clarify.

---

> ### Author Response · Authors · 2022-08-02
> **Response 1/2**
>
> $~$
>
> Thank you for your careful reading of our paper and your detailed questions. It appears the main concerns are i) resetting the covariance each episode, ii) unclear contribution compared to PC-PG and PC-MLP, and iii) the limited improvements in Vizdoom with image-based observations. We address each of these points below. The crucial point is that we are considering CMDPs, where at the start of every episode some aspects of the environment change, and that we compare to a wide range of published state-of-the-art baselines in that problem setting.
>
> $~$
>
> ### Limited improvements in image-based VizDoom
>
> The experiments on Vizdoom serve as a sanity check that our method can work with pixel-based observations, but it is not a CMDP - the environment/map is the same each episode. Since it is a singleton MDP, it is not too surprising that the baselines which are designed for singleton MDPs work well, and our method does not offer much improvement over them.
>
> However, since the initial submission we have performed additional experiments in reward-free exploration using the embodied AI simulator Habitat, which we believe is a much more interesting and considerably more difficult setting (please see Section 5.2 in the updated paper). In addition to having rich pixel-based observations and simulating real-world spaces, this is truly a CMDP because in each episode, the agent is initialized in a different simulated house (there are 1000 houses total, split into train/test sets). Here, E3B provides a much more significant improvement over existing methods. These results highlight the relative strength of our method over baselines in the more general (and more difficult) CMDP setting.
>
> $~$
>
> ### Resetting the covariance each episode
>
> First, we would like to again emphasize that our algorithm is designed for the contextual MDP framework, where each episode corresponds to a different environment, rather than the standard MDP framework, where the agent is spawned in the same environment each episode. This distinction is important for understanding why the covariance matrix is reset each episode.
>
> The maze example you describe (if we understand correctly) falls into the standard (singleton) MDP framework where the agent is spawned in the same maze each episode. Let’s assume for simplicity that the feature extractor $\phi$ extracts the $(x,y)$ position of the agent. In this case, indeed it does make sense to compute the covariance matrix across all of the previous episodes. In this way, if the agent has visited the left half of the maze over the course of previous episodes, then the covariance matrix will reflect this and the bonus will be low for the left half of the maze and high for the right half, which will encourage exploration of the unseen right half of the maze.
>
> However, an example which is closer to the contextual MDP framework we are considering would be where the maze is regenerated randomly at each episode (this is similar to MiniGrid/MiniHack tasks where the map layout is randomly generated each episode). In this case, if the agent has visited the left half of the maze at episode N, this doesn’t mean it shouldn’t visit the left half of the maze at episode N+1, because they are two different mazes. Since each episode corresponds to a distinct maze, we want our agent to learn a policy which explores as much as possible of the maze within each episode, and resetting the covariance matrix each episode encourages this.
>
> Also, observe that the count-based episodic bonus $N_e(s)$ used in RIDE, AGAC and NovelD resets after each episode, and our experiments in Section 3 show that it is an **essential** driver of performance. The way our algorithm resets the covariance matrix after each episode is analogous to the count-based episodic bonus resetting after each episode.
>
> Note that we include a comparison to a version of the algorithm where the covariance matrix is _not_ reset at each episode (this is denoted E3B (non-episodic) in Figure 7), and see a large drop in performance. **Our empirical results show  resetting the covariance matrix is indeed beneficial in the CMDPs we consider**.
>
> Based on our extensive experiments on MiniHack, VizDoom and Habitat (a total of 26 tasks, including ones with high-dimensional pixels), we believe that E3B in its current form is effective on a wide range of CMDPs. However, it is possible it would fail on certain problems such as the bidirectional diabolical combination lock described in the PC-PG paper (which is similar to the example you describe). Note however that other methods like RND, which use cross-episode novelty, also fail on this task. It is currently not clear to us how to incorporate ideas such as a policy cover used in singleton MDPs to the CMDP case, but this could be interesting future work.

---

### Author Response · Authors · 2022-08-02
**Paper revision with new results**

Thank you to all the reviewers for putting time and care into reviewing the paper. We answer individual questions and comments below. Furthermore, in response to Reviewer 1, we have added a revision of the paper which includes new results on a challenging photorealistic 3D environment (Habitat), providing additional evidence that E3B is effective in high dimensional pixel-based settings. While the Vizdoom results were a useful sanity check that E3B can work with pixels, the baselines were already able to solve it, so there was not much room for improvement. We believe this is because unlike MiniHack, Vizdoom is a singleton MDP where the environment is the same at each episode, so existing methods already work well.

We have added new experiments for reward-free exploration in the Habitat embodied AI simulator, which unlike Vizdoom, is a contextual MDP where the environment changes each episode (each episode, the agent finds itself in a different house). Here we see a large improvement in E3B’s performance over the baselines. We believe this gives further strong evidence that E3B is effective for exploring contextual MDPs (CMDPs) with rich, pixel-based observations, and reinforces E3B’s broad applicability.

Due to the 9 page limit for submissions, we moved the Vizdoom results to Appendix C.2, and also moved the figure in Section 4.1 to Appendix B to make space for the Habitat results. Since an additional page is allowed for the camera ready version, we will add these back to the main text if the paper is accepted.

We believe we have addressed the reviewers’ concerns and that, as a result, the paper has become much stronger.

---

### Meta-Review · Area_Chair_meDi · 2022-09-09

**Recommendation:** Accept
**Confidence:** Certain

**Metareview:**

The reviewers appreciated the fundamental questions the paper was asking, clear writing and argumentation of the paper and convincing empirical experiments. While there were concerns about the theoretical rationale of resetting covariance matrix, the empirical results show it is indeed important. For these reasons, I recommend acceptance.

**Award:**

No

---

### Decision · Program_Chairs · 2022-09-14

Accept